# Discover, Hallucinate, and Adapt: Open Compound Domain Adaptation for Semantic Segmentation

**Kwanyong Park, Sanghyun Woo, Inkyu Shin, In So Kweon**
Korea Advanced Institute of Science and Technology (KAIST)
{pkyong7,shwoo93,dlsrbgg33,iskweon77}@kaist.ac.kr

## Abstract

Unsupervised domain adaptation (UDA) for semantic segmentation has been attracting attention recently, as it could be beneficial for various label-scarce real-world scenarios (e.g., robot control, autonomous driving, medical imaging, etc.). Despite the significant progress in this field, current works mainly focus on a single-source single-target setting, which cannot handle more practical settings of multiple targets or even unseen targets. In this paper, we investigate open compound domain adaptation (OCDA), which deals with mixed and novel situations at the same time, for semantic segmentation. We present a novel framework based on three main design principles: *discover*, *hallucinate*, and *adapt*. The scheme first clusters compound target data based on style, discovering multiple latent domains (**discover**). Then, it hallucinates multiple latent target domains in source by using image-translation (**hallucinate**). This step ensures the latent domains in the source and the target to be paired. Finally, target-to-source alignment is learned separately between domains (**adapt**). In high-level, our solution replaces a hard OCDA problem with much easier multiple UDA problems. We evaluate our solution on standard benchmark GTA5 to C-driving, and achieved new state-of-the-art results.

## 1 Introduction

Deep learning-based approaches have achieved great success in the semantic segmentation [24, 43, 2, 7, 42, 3, 17, 10], thanks to a large amount of fully annotated data. However, collecting large-scale accurate pixel-level annotations can be extremely time and cost consuming [6]. An appealing alternative is to use off-the-shelf simulators to render synthetic data for which ground-truth annotations are generated automatically [33, 34, 32]. Unfortunately, models trained purely on simulated data often fail to generalize to the real world due to the *domain shifts*. Therefore, a number of unsupervised domain adaptation (UDA) techniques [11, 38, 1] that can seamlessly transfer knowledge learned from the label-rich source domain (*simulation*) to an unlabeled new target domain (*real*) have been presented.

Despite the tremendous progress of UDA techniques, we see that their experimental settings are still far from the real-world. In particular, existing UDA techniques mostly focus on a single-source single-target setting [37, 39, 45, 13, 25, 31, 5, 29]. They do not consider a more practical scenario where the target consists of multiple data distributions without clear distinctions. To investigate a continuous and more realistic setting for domain adaptation, we study the problem of open compound domain adaptation (OCDA) [23]. In this setting, the target is a union of multiple homogeneous domains without domain labels. The unseen target data also needs to be considered at the test time, reflecting the realistic data collection from both mixed and novel situations.

A naive way to perform OCDA is to apply the current UDA methods directly, viewing the compound target as a uni-modal distribution. As expected, this method has a fundamental limitation; It induces

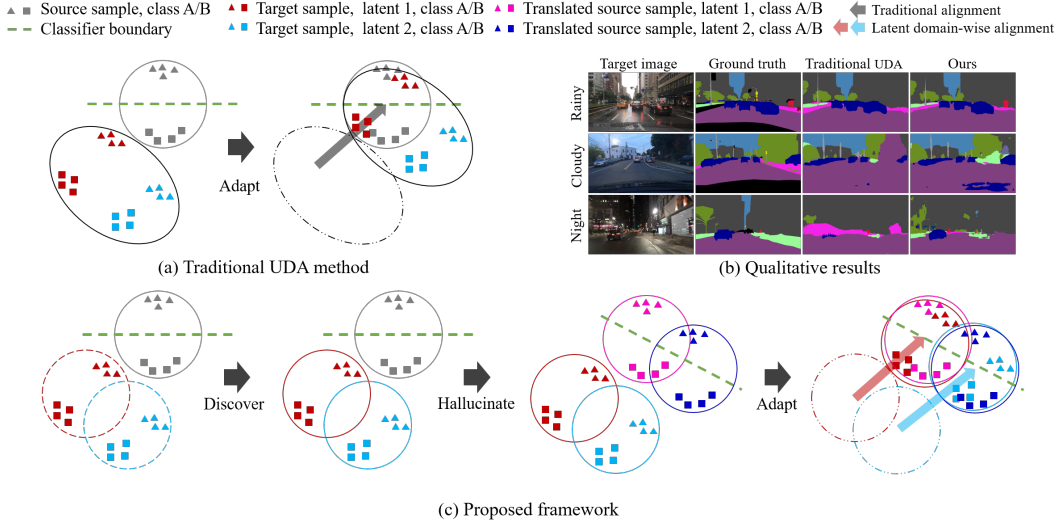

Figure 1: **Overview of the proposed OCDA framework: Discover, Hallucinate, and Adapt.** The traditional UDA methods consider compound target data as a uni-modal distribution and adapt it at once. Therefore, only the target data that is close to the source tends to align well (*biased alignment*). On the other hand, the proposed scheme explicitly finds multiple latent target domains and adopts domain-wise adversaries. The qualitative results demonstrates that our solution indeed resolves the biased-alignment issues successfully. We adopt AdaptSeg [37] as the baseline UDA method.

a *biased alignment*[1], where only the target data that are close to source aligns well (see Fig. 1 and Table 2-(b)). We note that the compound target includes various domains that are both close to and far from the source. Therefore, alignment issues occur if multiple domains and their differences in target are not appropriately handled. Recently, Liu *et.al.* [23] proposed a strong OCDA baseline for semantic segmentation. The method is based on easy-to-hard curriculum learning [45], where the easy target samples that are close to the source are first considered, and hard samples that are far from the source are gradually covered. While the method shows better performance than the previous UDA methods, we see there are considerable room for improvement as they do not fully utilize the domain-specific information[2].

To this end, we propose a new OCDA framework for semantic segmentation that incorporates three key functionalities: discover, hallucinate, and adapt. We illustrate the proposed algorithm in Fig. 1. Our key idea is simple and intuitive: decompose a hard OCDA problem into multiple easy UDA problems. We can then ease the optimization difficulties of OCDA and also benefit from the various well-developed UDA techniques. In particular, the scheme starts by discovering $K$ latent domains in the compound target data [28] (**discover**). Motivated by the previous works [15, 18, 26, 14, 4, 35] that utilizes style information as domain-specific representation, we propose to use *latent target styles* to cluster the compound target. Then, the scheme generates $K$ target-like source domains by adopting an examplar-guided image translation network [5, 40], hallucinating multiple latent target domains in source (**hallucinate**). Finally, the scheme matches the latent domains of source and target, and by using $K$ different discriminators, the domain-invariance is captured separately between domains [37, 39] (**adapt**).

We evaluate our framework on standard benchmark, GTA5 [33] to C-driving, and achieved new state-of-the-art OCDA performances. To empirically verify the efficacy of our proposals, we conduct extensive ablation studies. We confirm that three proposed design principles are complementary to each other in constructing an accurate OCDA model.

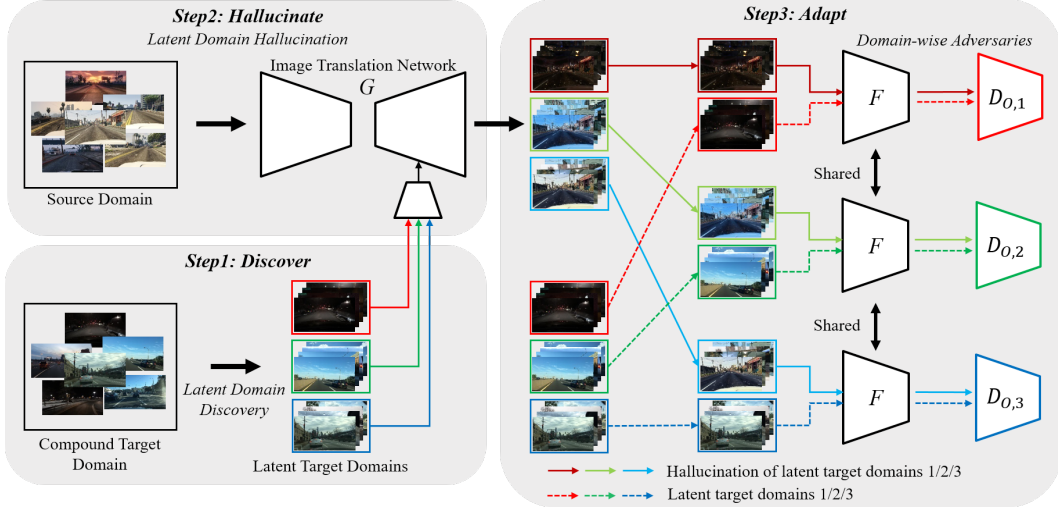

Figure 2: **Overview of the proposed network.** Following the proposed DHA (**Discover**, **Hallucinate**, and **Adapt**) training scheme, the network is composed of three main blocks. 1) **Discover**: Regarding the 'style' as domain-specific representation, the network partitions the compound target data into a total of $K$ clusters. We see each cluster as a specific latent domain. 2) **Hallucinate**: In the source domain, the network hallucinates $K$ latent targets using image-translation method. The source images are then closely aligned with the target, reducing the domain gap in a pixel-level. 3) **Adapt**: The network utilizes $K$ different discriminators to enforce domain-wise adversaries. In this way, we are able to explicitly leverage the latent multi-mode structure of the data. Connecting all together, the proposed network successfully learns domain-invariance from the compound target.

## 2 Method

In this work, we explore OCDA for semantic segmentation. The goal of OCDA is to transfer knowledge from the labeled source domain $S$ to the unlabeled compound target domain $T$, so that trained model can perform the task well on both $S$ and $T$. Also, at the inference stage, OCDA tests the model in open domains that have been previously unseen during training.

### 2.1 Problem setup

We denote the source data and corresponding labels as $X_S = \left\{\mathbf{x}_S^i\right\}_{i=1}^{N_S}$ and $Y_S = \left\{\mathbf{y}_S^i\right\}_{i=1}^{N_S}$, respectively. $N_S$ is the number of samples in the source data. We denote the compound target data as $X_T = \left\{\mathbf{x}_T^i\right\}_{i=1}^{N_T}$, which are from the mixture of multiple homogeneous data distributions. $N_T$ is the number of samples in the compound target data. We assume that all the domains share the same space of classes (i.e., closed label set).

### 2.2 DHA: Discover, Hallucinate, and Adapt

The overview of the proposed network is shown in Fig. 2, which consists of three steps: Discover, Hallucinate, and Adapt. The network first discovers multiple latent domains based on style-based clustering in the compound target data (**Discover**). Then, it hallucinates found latent target domains in source by translating the source data (**Hallucinate**). Finally, domain-wise target-to-source alignment is learned (**adapt**). We detail each step in the following sections.

#### 2.2.1 Discover: Multiple Latent Target Domains Discovery

The key motivation of the discovery step is to make *implicit* multiple target domains *explicit* (see Fig. 1 (c) - Discover). To do so, we collect domain-specific representations of each target image and assign pseudo domain labels by clustering (*i.e.*, $k$-means clustering [16]). In this work, we assume that the latent domain of images is reflected in their **style** [15, 18, 26, 14, 4, 35], and we thus use style information to cluster the compound target domain. In practice, we introduce hyperparameter $K$

and divide the compound target domain $T$ into a total of $K$ latent domains by style, $\{T_j\}_{j=1}^K$. Here, the style information is the convolutional feature statistics (i.e., mean and standard deviations), following [14, 9]. After the discovery step, the compound target data $X_T$ is divided into a total of $K$ mutually exclusive sets. The target data in the $j$-th latent domain ($j \in 1, ..., K$), for example, can be expressed as following: $X_{T,j} = \left\{\mathbf{x}_{T,j}^i\right\}_{i=1}^{N_{T,j}}$, where $N_{T,j}$ is the number of target data in the $j$-th latent domain [3].

### 2.2.2  Hallucinate: Latent Target Domains Hallucination in Source

We now hallucinate $K$ latent target domains in the source domain. In this work, we formulate it as image-translation [22, 44, 15, 18]. For example, the hallucination of the $j$-th latent target domain can be expressed as, $G(\mathbf{x}_S^i, \mathbf{x}_{T,j}^z) \mapsto \mathbf{x}_{S,j}^i$. Where $\mathbf{x}_S^i \in X_S$, $\mathbf{x}_{T,j}^z \in X_{T,j}$, and $\mathbf{x}_{S,j}^i \in X_{S,j}$ [4] are original source data, randomly chosen target data in $j$-th latent domain, and source data translated to $j$-th latent domain. $G(\cdot)$ is exemplar-guided image-translation network. $z \in 1, ..., N_{T,j}$ indicates random index. We note that random selection of latent target data improves model robustness on (target) data scarcity.

Now, the question is how to design an effective image-translation network, $G(\cdot)$, which can satisfy all the following conditions at the same time. 1) high-resolution image translation, 2) source-content preservation, and 3) target-style reflection. In practice, we adopt a recently proposed exemplar-guided image-translation framework called TGCF-DA [5] as a baseline. We see it meets two former requirements nicely, as the framework is cycle-free [5] and uses a strong semantic constraint loss [13]. In TGCF-DA framework, the generator is optimized by two objective functions: $L_{GAN}$, and $L_{sem}$. We leave the details to the appendicies as they are not our novelty.

Despite their successful applications in UDA, we empirically observe that the TGCF-DA method cannot be directly extended to the OCDA. The most prominent limitation is that the method fails to reflect diverse target-styles (from multiple latent domains) to the output image and rather falls into mode collapse. We see this is because the synthesized outputs are not guaranteed to be style-consistent (i.e., the framework lacks style reflection constraints). To fill in the missing pieces, we present a *style consistency loss*, using discriminator $D_{Sty}$ associated with a pair of target images - either both from same latent domain or not:

$$
\begin{aligned}
L_{Style}^j(G, D_{Sty}) = \ &\mathbb{E}_{\mathbf{x}_{T,j}' \sim X_{T,j}, \mathbf{x}_{T,j}'' \sim X_{T,j}} \left[ log D_{Sty}(\mathbf{x}_{T,j}', \mathbf{x}_{T,j}'') \right] \\
&+ \sum_{l \neq j} \mathbb{E}_{\mathbf{x}_{T,j} \sim X_{T,j}, \mathbf{x}_{T,l} \sim X_{T,l}} \left[ log(1 - D_{Sty}(\mathbf{x}_{T,j}, \mathbf{x}_{T,l})) \right] \\
&+ \mathbb{E}_{\mathbf{x}_S \sim X_S, \mathbf{x}_{T,j} \sim X_{T,j}} \left[ log(1 - D_{Sty}(\mathbf{x}_{T,j}, G(\mathbf{x}_S, \mathbf{x}_{T,j}))) \right]
\end{aligned}
\tag{1}
$$

where $\mathbf{x}_{T,j}'$ and $\mathbf{x}_{T,j}''$ are a pair of sampled target images from same latent domain $j$ (i.e., same style), $\mathbf{x}_{T,j}$, and $\mathbf{x}_{T,l}$ are a pair of sampled target images from different latent domain (i.e., different styles). The discriminator $D_{Sty}$ learns awareness of style consistency between pair of images. Simultaneously, the generator G tries to fool $D_{Sty}$ by synthesizing images with the same style to exemplar, $\mathbf{x}_{T,j}$. With the proposed adversarial style consistency loss, we empirically verify that the target style-reflection is strongly enforced.

By using image-translation, the hallucination step reduces the domain gap between the source and the target in a pixel-level. Those translated source images are closely aligned with the compound target images, easing the optimization difficulties of OCDA. Moreover, various latent data distributions can be covered by the segmentation model, as the translated source data which changes the classifier boundary is used for training (see Fig. 1 (c) - Hallucinate).

### 2.2.3  Adapt: Domain-wise Adversaries

Finally, given $K$ target latent domains $\{T_j\}_{j=1}^{K}$ and translated $K$ source domains $\{S_j\}_{j=1}^{K}$, the model attempts to learn domain-invariant features. Under the assumption of translated source and latent targets are both a uni-modal now, one might attempt to apply the existing state-of-the-art UDA methods (*e.g.*Adaptseg [37], Advent [39]) directly. However, as the latent multi-mode structure is not fully exploited, we see this as sub-optimal and observe its inferior performance. Therefore, in this paper, we propose to utilize $K$ different discriminators, $D_{O,j}, j \in 1, ..., K$ to achieve (latent) domain-wise adversaries instead. For example, $j$-th discriminator $D_{O,j}$ only focuses on discriminating the output probability of segmentation model from $j$-th latent domain (i.e., samples either from $T_j$ or $S_j$). The adversarial loss for $j$th target domain is defined as:

$$L_{Out}^{j}(F, D_{O,j}) = \mathbb{E}_{\mathbf{x}_{S,j} \sim X_{S,j}} \left[ log D_{O,j}(F(\mathbf{x}_{S,j})) \right] + \mathbb{E}_{\mathbf{x}_{T,j} \sim X_{T,j}} \left[ log(1 - D_{O,j}(F(\mathbf{x}_{T,j}))) \right] \quad (2)$$

where $F$ is segmentation network. The (segmentation) task loss is defined as standard cross entropy loss. For example, the source data translated to the $j$-th latent domain can be trained with the original annotation as:

$$L_{task}^{j}(F) = -\mathbb{E}_{(\mathbf{x}_{S,j}, \mathbf{y}_S) \sim (X_{S,j}, Y_S)} \sum_{h,w} \sum_{c} y_s^{(h,w,c)} log(F(\mathbf{x}_{S,j}))^{(h,w,c)})) \quad (3)$$

We use the translated source data $\{X_{S,j}\}_{j=1}^{K}$ and its corresponding labels $Y_s$.

### 2.3  Objective Functions

The proposed DHA learning framework utilizes adaptation techniques, including pixel-level alignment, semantic consistency, style consistency, and output-level alignment. The overall objective loss function of DHA is:

$$L_{total} = \sum_{j} \left[ \lambda_{GAN} L_{GAN}^{j} + \lambda_{sem} L_{sem}^{j} + \lambda_{Style} L_{Style}^{j} + \lambda_{Out} L_{Out}^{j} + \lambda_{task} L_{task}^{j} \right] \quad (4)$$

Here, we use $\lambda_{GAN} = 1$, $\lambda_{sem} = 10$, $\lambda_{Style} = 10$, $\lambda_{out} = 0.01$, $\lambda_{task} = 1$. Finally, the training process corresponds to solving the following optimization, $F^* = \arg \min_F \min_D \max_G L_{total}$, where G and D represents a generator (in $L_{sem}$, $L_{GAN}$, and $L_{Style}$) and all the discriminators (in $L_{GAN}$, $L_{Style}$, and $L_{Out}$), respectively.

## 3  Experiments

In this section, we first introduce experimental settings and then compare the segmentation results of the proposed framework and several state-of-the-art methods both quantitatively and qualitatively, followed by ablation studies.

### 3.1  Experimental Settings

**Datasets.**  In our adaptation experiments, we take GTA5 [33] as the source domain, while the BDD100K dataset [41] is adopted as the compound ("rainy", "snowy", and "cloudy") and open domains ("overcast") (*i.e.*, C-Driving [23]).

**Baselines.**  We compare our framework with the following methods. **(1) Source-only,** train the segmentation model on the source domains and test on the target domain directly. **(2) UDA methods,** perform OCDA via (single-source single-target) UDA, including AdaptSeg [37], CBST [45], IBN-Net [30], and PyCDA [21].  **(3) OCDA method,** Liu *et.al.* [23], which is a recently proposed curriculum-learning based [45] strong OCDA baseline.

Table 1: **Comparison with the state-of-the-art UDA/OCDA methods and Ablation study on framework design.** We evaluate the semantic segmentation results, GTA5 to C-driving. (a) † indicates the models trained on a longer training scheme. (b) "+trad" denote adopting traditional unsupervised method [37]

(a) Comparison with the state-of-the-art UDA/OCDA methods

| Source | Compound(C) | | | Open(O) | Avg. | |
|---|---|---|---|---|---|---|
| GTA5 | Rainy | Snowy | Cloudy | Overcast | C | C+O |
| Source Only | 16.2 | 18.0 | 20.9 | 21.2 | 18.9 | 19.1 |
| AdaptSeg [37] | 20.2 | 21.2 | 23.8 | 25.1 | 22.1 | 22.5 |
| CBST [45] | 21.3 | 20.6 | 23.9 | 24.7 | 22.2 | 22.6 |
| IBN-Net [30] | 20.6 | 21.9 | 26.1 | 25.5 | 22.8 | 23.5 |
| PyCDA [21] | 21.7 | 22.3 | 25.9 | 25.4 | 23.3 | 23.8 |
| Liu *et.al.* [23] | 22.0 | 22.9 | 27.0 | 27.9 | 24.5 | 25.0 |
| Ours | 27.0 | 26.3 | 30.7 | 32.8 | **28.5** | **29.2** |
| Source only† | 23.3 | 24.0 | 28.2 | 30.2 | 25.7 | 26.4 |
| Ours† | 27.1 | 30.4 | 35.5 | 36.1 | **32.0** | **32.3** |

(b) Ablation study on framework design.

| Method | Discover | Hallucinate | Adapt | C | C+O |
|---|---|---|---|---|---|
| Source Only | | | | 25.7 | 26.4 |
| Traditional UDA [37] | | | +trad | 28.8 | 29.3 |
| (1) | ✓ | | ✓ | 31.1 | 31.1 |
| (2) | ✓ | ✓ | | 29.8 | 30.4 |
| (3) | ✓ | ✓ | +trad | 30.1 | 31.0 |
| Ours | ✓ | ✓ | ✓ | **32.0** | **32.3** |

**Evaluation Metric.** We employ standard mean intersection-over-union (mIoU) to evaluate the segmentation results. We report both results of individual domains of compound("rainy", "snowy", "cloudy") and open domain("overcast") and averaged results.

**Implementation Details.**

- **Backbone** We use a pre-trained VGG-16 [36] as backbone network for all the experiments.

- **Training** By design, our framework can be trained in an end-to-end manner. However, we empirically observe that splitting the training process into two steps allows stable model training. In practice, we cluster the compound target data based on their style statistics (we use ImageNet-pretrained VGG model [36]). With the discovered latent target domains, we first train the hallucination step. Then, using both the translated source data and clustered compound target data, we learn the target-to-source adaptation. We adopt two different training schemes (short and long) for the experiments. For the short training scheme (5K iteration), we follow the same experimental setup of [23]. For the longer training scheme (150K iteration), we use LS GAN [27] for Adapt-step training.

- **Testing** We follow the conventional inference setup [23]. Our method shows superior results against the recent approaches without any overhead in test time.

## 3.2 Comparison with State-of-the art

We summarize the quantitative results in Table 1. we report adaptation performance on GTA5 to C-Driving. We compare our method with Source-only model, state-of-the-art UDA-models [37, 45, 30, 21, 39], and recently proposed strong OCDA baseline model [23]. We see that the proposed DHA framework outperforms all the existing competing methods, demonstrating the effectiveness of our proposals. We also provide qualitative semantic segmentation results in Fig. 3. We can observe clear improvement against both source only and traditional adaptation models [37].

We observe adopting a longer training scheme improves adaptation results († in Table 1 indicates models trained on a longer training scheme). Nevertheless, our approach consistently brings further improvement over the baseline of source-only, which confirms its enhanced adaptation capability. Unless specified, we conduct the following ablation experiments on the longer-training scheme.

## 3.3 Ablation Study

We run an extensive ablation study to demonstrate the effectiveness of our design choices. The results are summarized in Table 1-(b) and Table 2. Furthermore, we additionally report the night domain adaptation results (We see the night domain as one of the representative latent domains that are distant from the source).

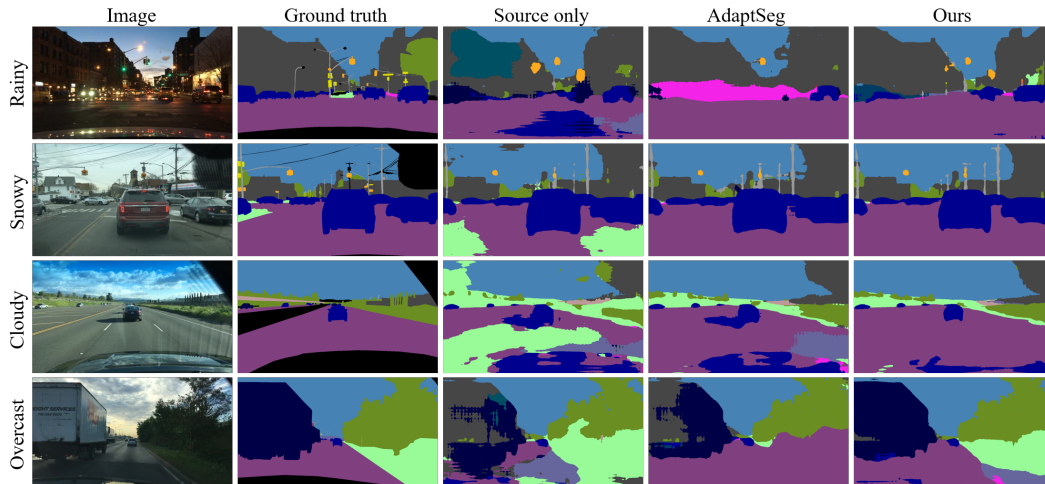

|  | Image | Ground truth | Source only | AdaptSeg | Ours |
|--|-------|--------------|-------------|----------|------|
| Rainy | | | | | |
| Snowy | | | | | |
| Cloudy | | | | | |
| Overcast | | | | | |

Figure 3: **Qualitative results comparison** of semantic segmentation on the compound domain("rainy", "snowy", "cloudy") and open domain("overcast"). We can observe clear improvement against both source only and traditional adaptation models [37].

**Framework Design.** In this experiment, we evaluate three main design principles: *Discover*, *Hallucinate*, and *Adapt*. We set the adaptation results of both Source Only and Traditional UDA [37] as baselines. First, we investigate the importance of Discover stage (Method (1) in Table 1-(b)). The method-(1) learns target-to-source alignment for each clustered latent target domain using multiple discriminators. As improved results indicate, explicitly clustering the compound data and leveraging the latent domain information allows better adaptation. Therefore, we empirically confirm our '*cluster-then-adapt*' strategy is effective. We also explore the Hallucination stage (Method (2) and (3) in Table 1-(b)). The method-(2) can be interpreted as a strong Source Only baseline that utilizes translated target-like source data. The method-(3) further adopts traditional UDA on top of it. We see both (2) and (3) outperform Source Only and Traditional UDA adaptation results, showing that hallucination step indeed reduces the domain gap. By replacing the Traditional UDA in method-(3) with the proposed domain-wise adversaries (Ours in Table 1-(a)), we achieve the best result. The performance improvement of our final model over the baselines is significant. Note, the final performance drops if any of the proposed stages are missing. This implies that the proposed three main design principles are indeed complementary to each other.

**Effective number of latent target domains.** In this experiment, we study the effect of latent domain numbers ($K$), a hyperparameter in our model. We summarize the ablation results in Table 2-(a). We vary the number of $K$ from 2 to 5 and report the adaptation results in the Hallucination Step. As can be shown in the table, we note that all the variants show better performance over the baseline (implying that the model performance is robust to the hyperparameter $K$), and the best adaptation results are achieved with $K = 3$. The qualitative images of found latent domains are shown in Fig. 4-(a). We can observe that the three discovered latent domains have their own 'style.' Interestingly, even these styles (e.g., $T_1$: night, $T_2$: clean blue, $T_3$: cloudy) do not exactly match the original dataset styles (e.g., "rainy", "snowy", "cloudy"), adaptation performance increases significantly. This indicates there are multiple implicit domains in the compound target by nature, and the key is to find them well and properly handling them. For the following ablation study, we set $K$ to 3.

**Style-consistency loss.** If we drop the style consistency loss in the hallucination step, our generator degenerates to the original TGCF-DA [5] model. The superior adaptation results of our method over the TGCF-DA [5] in Table 2-(a) implicitly back our claim that the target style reflection is not guaranteed on the original TGCF-DA formulation while ours does. In Fig. 4-(b), we qualitatively compare the translation results of ours and TGCF-DA [5]. We can obviously observe that the proposed style-consistency loss indeed allows our model to reflect the correct target styles in the output. This implies that the proposed solution enforces strong target-style reflection constraints effectively.

**Domain-wise adversaries.** Finally, we explore the effect of the proposed domain-wise adversaries in Table 2-(b). We compare our method with the UDA approaches, which consider both the translated

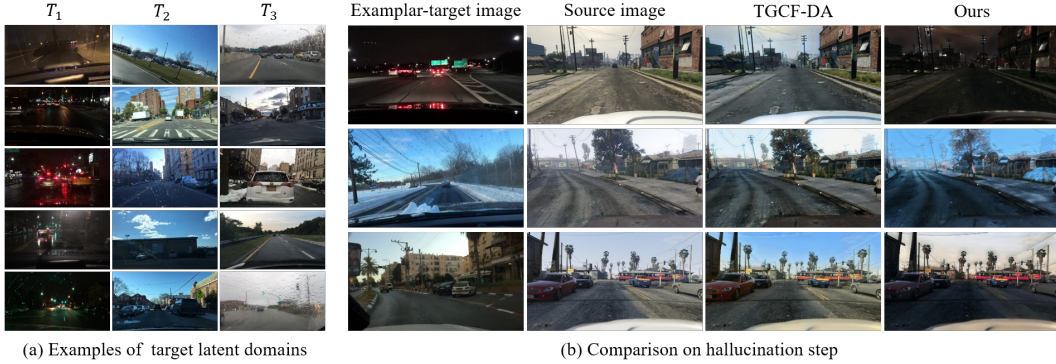

| $T_1$ | $T_2$ | $T_3$ | Examplar-target image | Source image | TGCF-DA | Ours |

(a) Examples of target latent domains          (b) Comparison on hallucination step

Figure 4: **Examples of target latent domains and qualitative comparison on hallucination step.** (a) We provide random images from each three latent domain (i.e., $K = 3$). Note that they have their own 'style.' (b) We show the effect of proposed style-consistency loss by comparing ours with original TGCF-DA [5] method.

Table 2: **(a)Ablation Study on the Discovery and Hallucination Step.** We conduct parameter analysis on $K$ to decide the optimal number of latent target domains. Also, we empirically verify the effectiveness of the proposed $L_{Style}$, outperforming TGCF-DA [5] significantly. (b)**Ablation Study on the Adapt step.** We confirm the efficacy of the proposed domain-wise adaptation, demonstrating its superior adaptation results over the direct application of UDA methods [37, 39].

(a) Discovery and Hallucination Step

| Source | Compound(C) | | | | Open(O) | Avg. | |
|---|---|---|---|---|---|---|---|
| GTA5 | Rainy | Snowy | Cloudy | Night | Overcast | C | C+O |
| Source Only | 23.3 | 24.0 | 28.2 | 8.1 | 30.2 | 25.7 | 26.4 |
| TGCF-DA [5] | 25.5 | 24.9 | 30.7 | 9.7 | 32.9 | 27.8 | 28.5 |
| Ours(K=2) | 26.0 | 26.6 | 32.4 | 11.1 | 33.6 | 29.3 | 29.7 |
| Ours(K=3) | **26.4** | **27.5** | **33.3** | 11.8 | **34.3** | **29.8** | **30.4** |
| Ours(K=4) | 25.2 | 26.4 | 32.7 | 12.1 | 33.8 | 29.1 | 29.5 |
| Ours(K=5) | 25.4 | 27.0 | 32.5 | **13.3** | 33.1 | 29.2 | 29.5 |

(b) Adapt Step

| Source | Adapt | Compound(C) | | | | Open(O) | Avg. | |
|---|---|---|---|---|---|---|---|---|
| | | Rainy | Snowy | Cloudy | Night | Overcast | C | C+O |
| Ours | None | 26.4 | 27.5 | 33.3 | 11.8 | 34.3 | 29.8 | 30.4 |
| Ours | Traditional( [37]) | 25.8 | 29.2 | 33.3 | 11.5 | 35.9 | 30.1 | 31.0 |
| Ours | Traditional( [39]) | 26.7 | 28.9 | 34.7 | 12.9 | 34.9 | 31.2 | 31.3 |
| Ours | Domain-wise( [37]) | 27.1 | 30.4 | 35.5 | 12.4 | 36.1 | **32.0** | **32.3** |
| Ours | Domain-wise( [39]) | 27.6 | 30.6 | 35.5 | 14.0 | 36.3 | **32.2** | **32.5** |

source and compound target as uni-modal and thus do not consider the multi-mode nature of the compound target. While not being sensitive to any specific adaptation methods (i.e., different UDA approaches such as Adaptseg [37] or Advent [39]), our proposal consistently shows better adaptation results over the UDA approaches. This implies that leveraging the latent multi-mode structure and conducting adaptation for each mode can ease the complex one-shot adaptation of compound data.

## 3.4 Further Analysis

**Quantitative Analysis on Biased Alignment.** In Fig. 1, we conceptually show that the traditional UDA methods induce *biased alignment* on the OCDA setting. We back this claim by providing quantitative results. We adopt two strong UDA methods, AdaptSeg [37] and Advent [39] and compare their performance with ours in GTA5 [33] to the C-driving [23]. By categorizing the target data by their attributes, we analyze the adaptation performance in more detail. In particular, we plot the performance/iteration for each attribute group separately.

We observe an interesting tendency; With the UDA methods, the target domains close to the source are well adapted. However, in the meantime, the adaptation performance of distant target domains are compromised [6]. In other words, the easy target domains dominate the adaptation, and thus the hard target domains are not adapted well (i.e., biased-alignment). On the other hand, the proposed DHA framework explicitly discovers multiple latent target domains and uses domain-wise adversaries

Figure 6: **Biased-alignment of UDA methods on OCDA.** The following graphs include testing mIoUs of traditional UDA methods [37, 39] and ours on GTA5 to C-driving setting. Note that the UDA methods [37, 39] tend to induce biased-alignment, where the target domains close to the source are mainly considered for adaptation. As a result, the performance of distant target domains such as "dawn" and "night" drops significantly as iteration increases. On the other hand, our method resolves this issue and adapts both close and distant target domains effectively.

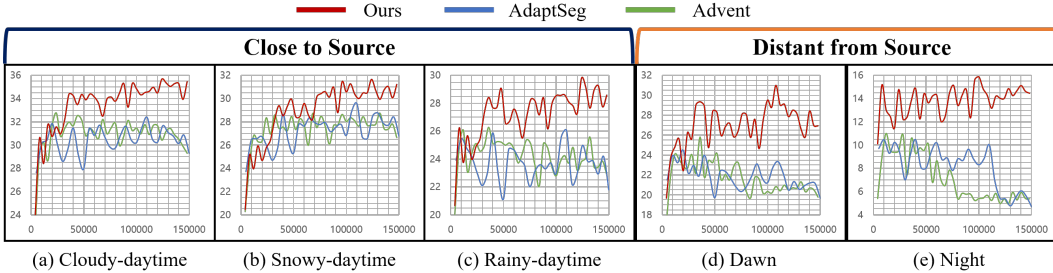

(a) Cloudy-daytime     (b) Snowy-daytime     (c) Rainy-daytime     (d) Dawn     (e) Night

to resolve the biased-alignment issue effectively. We can see that both the close and distant target domains are well considered in the adaptation (i.e., there is no performance drop in the distant target domains).

**Connection to Domain Generalization.** Our framework aims to learn domain-invariant representations that are robust on multiple latent target domains. As a result, the learned representations can well generalize on the unseen target domains (i.e., open domain) by construction. The similar learning protocols can be found in recent domain generalization studies [19, 28, 8, 20] as well.

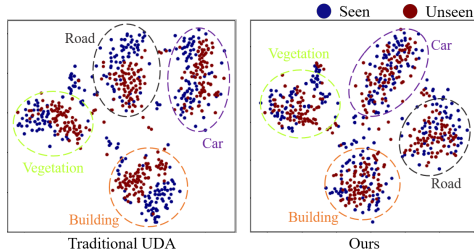

Figure 5: t-SNE visualization.

We analyze the feature space learned with our proposed framework and the traditional UDA baseline [39] in the Fig. 5. It shows that our framework yields more generalized features. More specifically, the feature distributions of seen and unseen domains are indistinguishable in our framework while not in traditional UDA [39].

# 4 Conclusion

In this paper, we present a novel OCDA framework for semantic segmentation. In particular, we propose three core design principles: Discover, Hallucinate, and Adapt. First, based on the latent target styles, we cluster the compound target data. Each group is considered as one specific latent target domain. Second, we hallucinate these latent target domains in the source domain via image-translation. The translation step reduces the domain gap between source and target and changes the classifier boundary of the segmentation model to cover various latent domains. Finally, we learn the target-to-source alignment domain-wise, using multiple discriminators. Each discriminator focuses only on one latent domain. Finally, we achieve to decompose OCDA problem into easier multiple UDA problems. Combining all together, we build a strong OCDA model for semantic segmentation. Empirically, we show that the proposed three design principles are complementary to each other. Moreover, the framework achieved new state-of-the-art OCDA results, outperforming the existing learning approaches significantly.

# Acknowledgements

This work was supported by Samsung Electronics Co., Ltd

## Broader Impact

We investigate the newly presented problem called open compound domain adaptation (OCDA). The problem well reflects the nature of real-world that the target domain often include mixed and novel situations at the same time. The prior work on this OCDA setting mainly focuses on the classification task. Though, we note that extending the classification model to the structured prediction task is non-trivial and requires significant domain-knowledge. In this work, we identify the challenges of OCDA in semantic segmentation and carefully design a new strong baseline model. Specifically, we present three core design principles: Discover, Hallucinate, and Adapt. We empirically show that our proposals are complementary to each other in constructing a strong OCDA model. We provide both the quantitative and qualitative results to show the efficacy of our final model. We hope the proposed new algorithm and its results will drive the research directions to step forward towards generalization in the real-world.

## Footnotes

[1]We provide quantitative analysis in Sec. 3.4.

[2]The OCDA formulation in [23] exploits domain-specific information. Though, it is only for the classification task, and the authors instead use a degenerated model for the semantic segmentation task as they cannot access the domain encoder. Please refer to the original paper for the details. This shows that extension of the framework from classification to segmentation (i.e., structured output) is non-trivial and requires significant domain knowledge.

[3] $X_{T,j}$ and $N_{T,j}$ satisfy $X_T = \bigcup_{j=1}^K X_{T,j}$ and $\sum_j N_{T,j} = N_T$, respectively.

[4] $X_{S,j} = \left\{\mathbf{x}_{S,j}^i\right\}_{i=1}^{N_S}$.

[5] Most existing GAN-based [12] image translation methods heavily rely on cycle-consistency [44] constraint. As cycle-consistency, by construction, requires redundant modules such as a target-to-source generator, they are memory-inefficient, limiting the applicability of high-resolution image translation.

[6]We see "cloudy-daytime", "snowy-daytime", and "rainy-daytime" as target domains close to the source, whereas "dawn" and "night" domain are distant target domains.

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
