[Supplementary Material]

# [Supplementary Material] Discover, Hallucinate, and Adapt: Open Compound Domain Adaptation for Semantic Segmentation

**Kwanyong Park, Sanghyun Woo, Inkyu Shin, In So Kweon**
Korea Advanced Institute of Science and Technology (KAIST)
{pkyong7,shwoo93,dlsrbgg33,iskweon77}@kaist.ac.kr

## A  Appendix

In this supplementary material, we provide more details about the model and experiments in the following order:

- In Sec. A.1, we evaluate our framework on two new datasets, Synscapes and SYNTHIA, demonstrating that our framework is general.

- In Sec. A.2, we conduct additional ablation studies on the adaptation step using four latent target domains (i.e., $K = 4$). We again see that the proposed domain-wise adversaries outperform the UDA approaches.

- In Sec. A.3, we analyze hyperparameter K selection.

- In Sec. A.4, we show more qualitative results.

- In Sec. A.5, we elaborate the implementation details.

Table 1: **Comparison with the state-of-the-art UDA methods.** We evaluate the semantic segmentation results, Synscapes [13] and SYNTHIA [8] to C-driving [6]. For SYNTHIA, we report averaged performance on 16 class subsets following the evaluation protocol used in [12, 15].

(a) Synscapes to C-driving

| Source Synscapes | Compound(C) | | | Open(O) | Avg. | |
|---|---|---|---|---|---|---|
| | Rainy | Snowy | Cloudy | Overcast | C | C+O |
| Source Only | 22.8 | 24.6 | 29.0 | 29.5 | 25.9 | 26.5 |
| CBST [15] | 23.1 | 25.1 | 30.1 | 30.0 | 26.5 | 27.0 |
| CRST [14] | 23.1 | 25.1 | 30.1 | 30.1 | 26.6 | 27.1 |
| AdaptSeg [11] | 24.2 | 26.2 | 31.6 | 31.2 | 27.9 | 28.3 |
| Advent [12] | 24.6 | 26.8 | 30.9 | 31.0 | 28.0 | 28.3 |
| Ours | 25.1 | 27.6 | 33.2 | 32.6 | **29.2** | **29.6** |

(b) SYNTHIA to C-driving

| Source Synscapes | Compound(C) | | | Open(O) | Avg. | |
|---|---|---|---|---|---|---|
| | Rainy | Snowy | Cloudy | Overcast | C | C+O |
| Source Only | 16.3 | 18.8 | 19.4 | 19.5 | 18.4 | 18.5 |
| CBST [15] | 16.2 | 19.6 | 20.1 | 20.3 | 18.9 | 19.1 |
| CRST [14] | 16.3 | 19.9 | 20.3 | 20.5 | 19.1 | 19.3 |
| AdaptSeg [11] | 17.0 | 20.5 | 21.6 | 21.6 | 20.0 | 20.2 |
| Advent [12] | 17.7 | 19.9 | 20.2 | 20.5 | 19.3 | 19.6 |
| Ours | 18.8 | 21.2 | 23.6 | 23.6 | **21.5** | **21.8** |

### A.1  DHA Framework on Other Datasets

We conduct OCDA semantic segmentation experiments using two additional benchmarks: Synscapes [13] and SYNTHIA [8]. We adopt the source-only method and the state-of-the-art UDA methods [11, 12, 15, 14] as baselines. The adaptation results are summarized in the Table 1. We observe that our method consistently outperforms previous UDA approaches on both datasets. This implies that our DHA framework is indeed general and practical for OCDA.

Table 2: **Ablation Study on the Adapt step.** The number of latent target domains are set to four (*i.e.*, $K = 4$). We again confirm the efficacy of the proposed domain-wise adaptation, demonstrating its superior adaptation results over the direct application of UDA methods [11, 12] in compound data.

| | | Compound(C) | | | | Open(O) | Avg. | |
|--------|------------------|-------|-------|--------|-------|----------|------|------|
| Source | Adapt | Rainy | Snowy | Cloudy | Night | Overcast | C | C+O |
| Ours | None | 25.2 | 26.4 | 32.7 | 12.1 | 33.8 | 29.1 | 29.5 |
| Ours | Traditional( [11]) | 25.4 | 28.3 | 33.5 | 10.8 | 34.7 | 29.7 | 30.5 |
| Ours | Traditional( [12]) | 25.9 | 27.8 | 34.2 | 10.6 | 34.7 | 30.1 | 30.7 |
| Ours | Domain-wise( [11]) | 24.6 | 28.8 | 35.0 | 12.0 | 35.1 | **30.7** | **30.9** |
| Ours | Domain-wise( [12]) | 26.7 | 29.9 | 34.8 | 13.5 | 35.8 | **31.4** | **31.8** |

## A.2 Additional Ablation Study on the Adapt Step

In the main paper, we already show that the proposed domain-wise adversaries are more effective than the traditional UDA approaches. To provide more experimental evidence, we conduct an additional ablation study using four latent target domains (i.e., $K = 4$). The results are shown in Table 2. We again observe that domain-wise adversaries show strong effectiveness compared to the traditional UDA approaches, confirming that explicitly leveraging the multi-mode nature of target data is essential. The tendency holds regardless of the UDA methods. We note that UDA methods in the night domain are even lower than the baseline, which can be interpreted as biased-alignment, as mentioned above. In contrast, the proposed method outperforms the baseline in every domain, achieving the best-averaged score.

## A.3 Analysis of the hyperparameter K Selection

If K value is much less than the optimal, the target distribution might be oversimplified, and some latent domains could be ignored. On the other hand, the images of similar styles might be divided into different clusters, and also each cluster may contain only a few images. In this work, we have set the value of K empirically. Instead, we see one can set the value using existing cluster evaluation metrics such as silhouette score [9]. It evaluates the resulting clusters by considering the intra-cluster variation and inter-cluster distance at the same time. As shown in the Fig. 1-(a), K=2 and 3 are the strong candidates, and the quality of clusters drops after K=3.

Figure 1: Silhouette score.

## A.4 Additional Qualitative Results

In Fig. 2, we provide more qualitative results.

## A.5 Implementation Details

Our model is implemented using Pytorch v0.4.1, CUDNN v7.6.5, CUDA v9.0.

**Discover step** We use ImageNet [3] pretrained Vgg-16 [10] to encode *style* of target images. Specificallly, we use relu1_2 features. All target images are resized to have width of 512 pixels while keeping the aspect ratio (*i.e.*, 512×288).

**Hallucination step** We detail the two objective functions, $L_{GAN}$ and $L_{sem}$, which are omitted in the main paper.

First, the $L_{GAN}$ [4] is defined as follows:

$$L_{GAN}^{j}(G, D_I) = \mathbb{E}_{\mathbf{x}_S \sim X_S, \mathbf{x}_{T,j} \sim X_{T,j}} log D_I(G(\mathbf{x}_S, \mathbf{x}_{T,j})) + \mathbb{E}_{\mathbf{x}_{T,j} \sim X_{T,j}} log [1 - D_I(\mathbf{x}_{T,j})] \quad (1)$$

Image discriminator $D_I$ learns to classify translated source and target images while the generator G tries to produce translated images that are visually similar to target images.

Second, to enforce strong semantic constraint, the $L_{sem}$ [5] is adopted in TGCF-DA [2] framework. It is defined as follows:

$$L_{sem}^{j}(G, f_{seg}) = - \mathbb{E}_{(\mathbf{x}_S, \mathbf{y}_S) \sim (X_S, Y_S), \mathbf{x}_{T,j} \sim X_{T,j}} \sum_{h,w} \sum_{c} \mathbf{y}_s^{(h,w,c)} log(f_{seg}(G(\mathbf{x}_S, \mathbf{x}_{T,j}))^{(h,w,c)}))$$

(2)

where $f_{seg}$ indicates the semantic segmentation model, which is pretrained on the labeled source domain. Weights of $f_{seg}$ are fixed during training. The loss function strongly encourages the model to preserve the semantics between the source image and the translated image.

In the hallucination step, the source and the target images are resized to 1280×720. For the memory-efficient training, we randomly crop the patches with a resolution of 1024×512. For the testing, we use the original size of 1280×720.

**Adapt step** We use segmentation model DeepLab V2 [1] (for the GTA5/Synscapes experiments) and FCN-8s [7] (for SYNTHIA experiments). As noted in the main paper, we use the VGG-16 backbone network. For the training, we resize the images of GTA5, Synscapes, and SYNTHIA to 1280×720, 1280×640, 1280×760, respectively [11, 12, 6]. We resize the target images in BDD100K to 960×540, following [6].

Figure 2: **Qualitative results.** We provide the semantic segmentation results on the compound domain ("rainy", "snowy", "cloudy") and open domain ("overcast"). We can observe clear improvement against both source only and traditional adaptation model [11].