[Reviews · NeurIPS 2020]

Review 1

Summary and Contributions: The authors extend the problem of unsupervised domain adaptation under open compound settings by proposing a three-stage pipeline, which leads to improved performance in the semantic segmentation task. More specifically, the proposed pipeline decomposes the OCDA problem into multiple UDA problems to achieve superior performance. Experiments under different settings demonstrate the effectiveness of the proposed approach.

Strengths: 1. The idea of solving the open compound domain adaptation problem with three different functionalities is interesting. 2. The paper is relatively well written and easy to follow. 3. A reasonable ablation study is provided to justify the different components used in their pipeline. 4. The generalization of the proposed domain-wise adaptation demonstrates superior performance over the baselines.

Weaknesses: 1. While the idea of jointly discovering, hallucinating, and adapting is interesting, there is a complete lack of discussing the impact of adding additional parameters and additional computational effort due to the multi-stage training and the multiple discriminators. The authors should provide this analysis for a fair comparison with the baseline [31, 33, *]. 2. Splitting the target data into easy and hard is already explored in the context of UDA. 3. Discovering the latent domain from the target domain is already proposed in [24]. 4. The problem of Open Compound Domain Adaptation is already presented in [**]. 5. Hallucinating the latent target domains is achieved through an image translation network adapted from [5]. 6. Style consistency loss to achieve diverse target styles has been used in previous works. 7. While the existing UDA methods [31,33] only use one discriminator, it is unclear to me why authors have applied multiple discriminators. 8. The details of the discriminator have not been discussed. 9. I was wondering why including the hallucination part reduces the performance in Table 1(b). It seems like the Discover module with [31] performs better than (Discover + Hallucinate + [31]). Also, the complex adapting stage where the authors used multiple discriminators mostly brings performance improvement. More importantly, did authors try to run the baseline models [17, 25, 31, 33, 39] with a similar longer training scheme? Otherwise, it is unfair to compare with the baselines. 10. Since the authors mentioned that splitting the training process helps to achieve better performance, It could be interesting to see the results of single-stage and multi-stage training. 11. It is not well explained why the adaptation performance drops when K > 3. Also, the procedure of finding the best K seems ad hoc and time-consuming. 12. I am just curious to see how the proposed method performs in a real domain adaptation scenario (GTA5->CityScapes). [*] Fei Pan, Inkyu Shin, François Rameau, Seokju Lee, In So Kweon. Unsupervised Intra-domain Adaptation for Semantic Segmentation through Self-Supervision. In CVPR 2020. [**] Liu, Ziwei and Miao, Zhongqi and Pan, Xingang and Zhan, Xiaohang and Lin, Dahua and Yu, Stella X. and Gong, Boqing. Open Compound Domain Adaptation. In CVPR 2020.

Correctness: Yes

Clarity: Decent

Relation to Prior Work: Yes

Reproducibility: Yes

Additional Feedback: While the idea of decomposing OCDA problem into multiple UDA problems is interesting in the context of semantic segmentation, the proposed method simply combines different existing components. The reported numbers in experiments suggest noticeable improvements; however, there are several issues mentioned in the weakness section that need to be explained before the final rating. After reading rebuttal information. ========================== As mentioned by all the reviewers (even the positive reviewer), this work is based on some prior works [31, 33, 39] despite the reasonable improvement. While I am a bit more positive about the paper after reading other reviewers' responses and other reviews, my original concern regarding the technical novelty and contributions persist. Interestingly, when the authors applied their proposed approach to a purely UDA setting (GTA5->Cityscapes), the improvement in terms of mIoU is not significant enough to be considered as an effective method. Additionally, some adhoc design choices (e.g., choosing the value of K) makes the approach a bit hacky which is agreed on by R2 and R4. So, I am not convinced the paper has enough novelty to be presented as a new contribution to the NeurIPS community. Therefore, I would like to keep my original score “Marginally below the acceptance threshold”. However, if the AC/other reviewers feel that this makes for a fundamental contribution in the area of unsupervised domain adaptation, then I'd be happy if the paper is accepted.


Review 2

Summary and Contributions: The paper presents a framework for open compound domain adaptation for semantic segmentation. The pipeline consists three steps. Firstly, the target data is spited into K cluster, each with a style. Then, the source domain is translated respectively into the K different styles. Lastly, domain invariance is enforced on the K target-source pairs by adversarial training. **after rebuttal** My initial concerns are mostly addressed in the rebuttal. Overall I found the framework interesting, and would recommend for acceptance. The experiments provided in the rebuttal should be added in the revision if accepted.

Strengths: - The proposed framework is clean and well-motivated, the empirical performance is better than the competing method[19]. - Ablation study is provided, which is helpful to access the individual contribution of each step.

Weaknesses: - I am not fully convinced by the effectiveness of the method, in a larger context. For example, the results are reported with only VGG backbones, it's not clear whether the proposed framework can be used to other backbones, such as ResNet which is used in most SOTA segmentation models. - Besides, the baselines used in table 1 are mainly UDA methods. It would be helpful to see if the proposed framework can be useful in a UDA setting (for example, GTAV->Cityscapes), and how the technique compare with existing methods in a UDA setting. It would be interesting to see if the method can provide some improvements to showcase the advantage of discovering latent domains. - I am also a bit skeptical of the selection of K. In my opinion, it might not be trivial to select K when there is no validation data available. However, the optimal K can be quite different depends on the data distribution. It would be helpful to comment on how to select K in practice. - To further understand how different components can help domain adaptation. I suggest to conduct tsne visualization on the feature distributions of different models. In particular, it would be interesting to see what's the distribution of the K clusters of the target data.

Correctness: Seems correct to me.

Clarity: The paper is well-written.

Relation to Prior Work: The contribution is clearly stated.

Reproducibility: Yes

Additional Feedback:


Review 3

Summary and Contributions: This paper proposes an open compound domain adaptation (OCDA) method for semantic segmentation, namely discover, hallucinate, and adapt (DHA). The idea is to cluster the data into several latent domains based on style (discover), transform the source domain into the latent domains (hallucinate), and apply adaptation from source to target separately for each domain. The proposed method achieves state-of-the-art results on the GTA to C-driving dataset. Ablation studies are conducted to evaluate the effectiveness of each component.

Strengths: - While part of the methodology borrows some existing methods (e.g., TGCF-DA, Adaptseg), overall the framework is novel and reasonable. And the problem studied (OCDA) has a more realistic setting than some prior domain adaptation tasks. - The experimental results are satisfactory. The proposed method shows notable improvements over its counterparts, achieving SOTA performance. Ablation studies are conducted to verify the effectiveness of the proposed components. - It is interesting to show that the latent domains discovered do not exactly match the original defined domains in the dataset. The discovered domains seem to have more apparent appearance characteristics.

Weaknesses: - The effectiveness of the ‘Hallucinate’ process is not validated. According to the ablation study results in Table 1 (b), the comparison between (1) and (3) suggests that the proposed ‘Hallucinate’ part does not improve the performance, but makes it worse. This result challenges the effectiveness and necessity of the Hallucinate process. I am wondering if remove this part from the framework would have no harm to the performance. - In my opinion, the illustration of ‘Hallucinate’ process in Fig.1 (c) is not very accurate. It should push the source domain closer towards the discovered latent domains. But this is not shown in the figure. - I am wondering why not conducting Hallucinate in the opposite direction, i.e., transform the latent domains towards the source domain, then apply adaptation between the original source domain and the transformed source domains. This would reduces the overall domain gaps and seems cleaner. Some discussions on this would be good. - For the style-consistency loss, authors have shown qualitative improvements. It would be better to also report quantitative improvements in the ablation study.

Correctness: Yes, the claims, method, and experiments are correct.

Clarity: The paper is generally well written. - Typo: Line 176: we->We, line 137: should x_s^’ be x_s,j?

Relation to Prior Work: Yes. The differences to prior works are clearly discussed.

Reproducibility: Yes

Additional Feedback: - See weakness. The main issue is that the effectiveness of the ‘Hallucinate’ process is not validated. - Some implementation details are missing, e.g., the learning rate and optimizer used for training. %%%%%%%%%%%%%%%%%%%%%%%%%%%%%%%%%%%%% After rebuttal period: After reading the rebuttal, I still feel that this submission has its merits to be accepted: 1) Though several components share similar spirits to prior works [31, 33, ...], the proposed framework is reasonable, clear and effective overall. Specifically, the idea of "discovering" and "hallucinating" latent domains, in the context of domain adapted segmentation, pushes one step further into tackling a more realistic OCDA setting [19]. 2) The manuscript is generally well-written and demonstrates its main contributions in an easy-to-follow manner. 3) Some initial concerns on the empirical evaluation (e.g. ablation study, various datasets and backbones, and the selection of K) have mostly been addressed in the rebuttal. Therefore, I would like to keep my original score as "A good submission; accept".


Review 4

Summary and Contributions: This paper presents a new method to address a new domain adaptation problem named open compound domain adaptation (OCDA), which emerges since CVPR 2020 [19]. The proposed method first discovers several clustering sub-domains in the target domain, then leverages image-to-image translation method to generate style-transferred source domain images, and then uses them together with the target domain images to carry out the adaptation process. Experiments on the benchmark datasets are carried out to validate the effectiveness of the proposed method.

Strengths: 1) This paper investigates the OCDA problem for semantic segmentation, which is an interesting and practical problem. The proposed DHA method provides a strong baseline for this task. 2) Assuming the target data obey multi-modal distributions, DHA separates target images into several sub-domains and perform appearance-level alignment by image translation. The idea seems interesting and reasonable. 3) An adjusted loss function, namely, style-consistent loss, is involved in exemplar-guided TGCF-DA to improve the domain-wise translation. 4) Extensive experimental results show effectiveness.

Weaknesses: 1) Authors claim that DHA can deal with the previously unseen target images (open domain) at the testing time but lacks convincing analysis. According to the “discover” and “hallucinate” steps, the image-to-image (I2I) translation model indeed learns to translate source domain images to have the same styles with the sub-domains identified in the target domain during the training stage. However, when applying the proposed model to unseen novel sub-domains in testing set (such as ‘overcast’ in C-driving), are the images from the unseen domains directly fed into the adaptation model? If so, how the adaptation model could adapt to these images with unseen novel styles? Does the adaptation model learn the generalization and adaptation ability despite the styles? The authors could present a T-SNE analysis to visualize the category features from the adaptation model for images with different styles. The authors need to further explain why an improvement sees in the open domain. Otherwise, it will doubt why DHA can handle the open domain and the paper is somewhat overclaimed. 2) Besides, prior UDA methods [A-B] also adopt the I2I technique as a pre-processing step to reduce the domain gap at the appearance level, although the typical paradigm is translating the source domain images to have the same styles as the target domain. Nevertheless, a strong baseline model could be established by following the “translating and adaptation” paradigm, where target domain images are translated into the source domains and the domain adaptation part could be replaced with state-of-the-art methods like [C-D]. In this way, it can also deal with unseen novel sub-domains in the testing set. A comparison with such baselines can better demonstrate the effectiveness and superiority of the proposed method. [A]. Bidirectional Learning for Domain Adaptation of Semantic Segmentation, in CVPR 2019. [B]. All about structure: Adapting structural information across domains for boosting semantic segmentation, in CVPR 2019. [C]. Category anchor-guided unsupervised domain adaptation for semantic segmentation, in NeurIPS 2019. [D]. Domain adaptation for semantic segmentation with maximum squares loss, in CVPR 2019. 3) The paper is somewhat hard to follow. As OCDA is an emerging task initially appearing in CVPR2020, authors should give a full explanation about the OCDA setting to make the paper clear. Besides, the analysis in the "Discover" step is not clear: (1) which layer is used to calculate the statistics of the features (mean and std) for clustering? (2) the number of images in each latent domain should be provided in terms of different hyper-parameters K. (3) according to Table 2 (a), the average performance peaks at K=3. Why the performance drops after K becomes larger? Is that because a small number of images in some sub-domain, leading to translated images with limited quality? Quantitative analysis on clustering results may answer the questions (e.g. intra-class and inter-class distances). 4) Are “Hallucinate” and “Adapt” steps trained end-to-end? Or Adapt is conducted after image generation finishes? 5) I notice that images are resized during training. Whether the original sizes are used for testing?

Correctness: Not aways. Further clarification and more empirical evidence are needed.

Clarity: The writting could be improved. Certain parts should be clarified further.

Relation to Prior Work: Not enough. Some state-of-the-art UDA methods can be included for comparison and discussion.

Reproducibility: No

Additional Feedback: The authors have addressed most of my concerns. Quantitative evaluation based on the clustering metric silhouette score is presented to show how to choose the hyper-parameter k. The t-sne results demonstrate that better feature alignment has been achieved by the proposed method than the baseline method. Although the performance on a typical UDA setting (GTA5->Cityscapes) is only marginally better than the baseline and falls behind other SOTA methods, the propose method can be considered as a new exploration in the OCDA setting. Therefore, I'd like to raise my score to "6: Marginally above the acceptance threshold".

[Author Response · NeurIPS 2020]

We thank all the reviewers for their constructive reviews. We answer each question below.

**Novelty & Contribution** We carefully design a unified OCDA framework for semantic segmentation. While some components adopt existing methods, these are well combined in a novel (task-specific) way as also noted by [R3]. We provide extensive ablation studies to verify the individual contribution of three complementary principles. We finally

(a) Clustering Evaluation      (b) T-SNE Visualization

Figure 1: Additional analysis.

achieved new state-of-the-art OCDA performance. We believe our findings and results can benefit the communities and practitioners.

**R1: Additional cost of adopting multi-stage training and multiple discriminators** Compared to the baselines, our DHA framework slightly increases the memory usage and computation at the training time. Specifically, the total training time of [31], [33], [*], and ours are 33.8hr, 34.1hr, 61.2hr, and 64.7hr, respectively. The according final performances are 28.8, 29.1, 29.5, and 32.0. This implies that our framework brings significant performance improvement with the moderate computational cost increase during training. We note that the test time costs are all the same, as we only utilize an identical segmentation model.

**R1: Why multiple discriminators?** To explicitly capture the underlying multi-mode structures in the data, we adopt using multiple discriminators. Our strong empirical results backs our design choice.

**R1,R3: Effectiveness of the hallucination step (Table 1-(b))** We apologize for the incorrect notations in the main paper Table 1-(b). As noted in the main paper (see section 3.3 Framework Design), we learn target-to-source alignment using multiple discriminators in Method-(1). Thus, the '+trad' must be replaced with /check. In fact, to see the effect of hallucination step, we should compare the result of the source only and Method-(2) or traditional UDA and Method-(3). The clear improvement demonstrates its efficacy.

**R1: Baselines with longer training scheme** The followings are the results: [Ours 32.0 / ADVENT 29.1 / Adaptseg 28.8 / CRST 26.9 / CBST 26.7 / Source-only 25.7]. Our framework acheives the best result. We note that the result of [19] is not included, since the official code (for semantic segmentation) is not available currently.

**R1,R4: end-to-end training** The end-to-end training causes the model to diverge.

**R1,R2,R4: Issues in the hyperparameter K** If K value is much less than the optimal, the target distribution might be oversimplified, and some latent domains could be ignored. On the other hand, the images of similar styles might be divided into different clusters, and also each cluster may contain only a few images. In this work, we have set the value of K empirically. Instead, we see one can set the value using existing cluster evaluation metrics such as silhouette score. It evaluates the resulting clusters by considering the intra-cluster variation and inter-cluster distance at the same time. As shown in the Fig. 1-(a), K=2 and 3 are the strong candidates, and the quality of clusters drops after K=3.

**R1,R2: Applying DHA framework on UDA setting (GTA5 to Cityscapes)** The followings are the results: [Ours 36.7 / ADVENT 36.1 / Cycada 35.4 / Adaptseg 35.0 / CBST 30.9]. Our framework achieves the best result.

**R2: DHA framework with the ResNet backbone** The followings are the results: [Ours 37.2 / ADVENT 36.0 / Adaptseg 36.2 / CRST 36.4/ CBST 35.8 / Source-only 35.7]. We achieve state-of-the-art again.

**R2,R4: T-SNE visualization** We analyze the feature space learned with our proposed framework and the advent baseline in the Fig. 1-(b). In appears that our framework yields more generalized features. More specifically, the feature distributions of seen and unseen domains are indistinguishable in our framework while not in advent.

**R3: Hallucinate in opposite direction** We rather observe degraded performance due to the undesirable translation. It is mainly because the semantic labels do not exist and the styles are diverse in the target domain.

**R3: Quantitative analysis on style consistency loss** In the main paper Table 2-(a), we already provided the quantitative ablation results (ours vs. TGCF-DA).

**R3: Figure 1-(c) modification** As suggested, we will modify the figure in the final version.

**R1,R3,R4: Missing implementation details** For the fair comparison, we use same discriminator, learning rate, optimizer, and train/test-time image resolutions with [31,33]; To compute the feature statistics in the "Discover" step, we use vgg-16 relu1_2.

**R4: How the DHA framework can deal with the open domain?** Our framework aims to learn domain-invariant representations that are robust on multiple latent target domains. As a result, the learned representations can well generalize on the unseen target domains by construction (please also refer to the Fig. 1-(b)). The similar learning protocols can be found in recent domain generalization studies as well.

**R4: Comparison with the another strong baseline ([A-B] + [C-D])** As mentioned above, hallucinating the images in the opposite direction produces undesirable images. Therefore, even with the strong translation technique of [A], we get inferior result (27.3) compared to the source only (35.7). Not surprisingly, applying the recent adaptation method of [D] on top of this translation result did not improve up to ours (A+D 30.3 / Ours 37.2).

[Meta-Review · NeurIPS 2020]

Four experts reviewed the paper. Three of them put the paper above the acceptance threshold, and one placed it marginally below. The reviewers were satisfied with the rebuttal in general, though R1 remained concerned about the technical novelty. Based on the reviewers' overall feedback, the decision is to recommend the paper for acceptance. The authors are encouraged to make the necessary changes to address the reviewers' questions to the best of their ability. Additionally, AC suggested reporting the results of both K=2 and K=3 due to their high silhouette scores. We congratulate the authors on the acceptance of their paper!